# Wearable Smart Textiles for Long-Term Electrocardiography Monitoring—A Review

**DOI:** 10.3390/s21124174

**Published:** 2021-06-17

**Authors:** Abreha Bayrau Nigusse, Desalegn Alemu Mengistie, Benny Malengier, Granch Berhe Tseghai, Lieva Van Langenhove

**Affiliations:** 1Department of Materials, Textiles and Chemical Engineering, Ghent University, 9000 Gent, Belgium; Benny.Malengier@UGent.be (B.M.); GranchBerhe.Tseghai@UGent.be (G.B.T.); lieva.vanlangenhove@ugent.be (L.V.L.); 2Ethiopian Institute of Textile and Fashion Technology, Bahir Dar University, Bahir Dar 6000, Ethiopia; dmengist@calpoly.edu; 3Materials Engineering Department, California Polytechnic State University, San Luis Obispo, CA 93407, USA

**Keywords:** dry electrode, electrocardiography, smart textiles, textile, textile electrode

## Abstract

The continuous and long-term measurement and monitoring of physiological signals such as electrocardiography (ECG) are very important for the early detection and treatment of heart disorders at an early stage prior to a serious condition occurring. The increasing demand for the continuous monitoring of the ECG signal needs the rapid development of wearable electronic technology. During wearable ECG monitoring, the electrodes are the main components that affect the signal quality and comfort of the user. This review assesses the application of textile electrodes for ECG monitoring from the fundamentals to the latest developments and prospects for their future fate. The fabrication techniques of textile electrodes and their performance in terms of skin–electrode contact impedance, motion artifacts and signal quality are also reviewed and discussed. Textile electrodes can be fabricated by integrating thin metal fiber during the manufacturing stage of textile products or by coating textiles with conductive materials like metal inks, carbon materials, or conductive polymers. The review also discusses how textile electrodes for ECG function via direct skin contact or via a non-contact capacitive coupling. Finally, the current intensive and promising research towards finding textile-based ECG electrodes with better comfort and signal quality in the fields of textile, material, medical and electrical engineering are presented as a perspective.

## 1. Introduction

With the rapid development in technology and the ever-increasing demands of people, conventional textiles are becoming inadequate for our uses. Traditionally, textile clothing is expected to have a good fit, comfort, and durability for use. Nowadays, these requirements are not enough due to growing competition on the market and changes in society demand supported with technological advancements. These societal demands and technological advancements led to the development of “smart textiles” [1]. Smart textiles are defined as materials that are able to change their behavior as a response to the influence of external factors or stimuli from the surrounding environment such as from mechanical, thermal, chemical, electrical, magnetic, or other sources [1,2]. Based on their level of “smartness”, smart textiles can be categorized into three subgroups: passive, active and very smart [2,3]. Passive smart textiles: textiles that only sense the environmental condition, and react to the stimuli passively, e.g., biopotential sensors. Active smart textiles: textiles that are able to sense the stimuli from the environment and respond to that particular stimulus. This can be achieved by integrating an actuator function and a sensing device. An example is a temperature-aware shirt which will automatically roll up its sleeves when body temperature becomes elevated. Very smart or intelligent textiles: textiles that are able to sense the environmental stimuli, give reaction to the stimuli, and thirdly adapt their behavior to the given circumstances. In the future, intelligent fabrics are expected to be integrated with cloud computing. For example, patients with a homecare medical device could send vital signals to their doctor to diagnose their health condition [4,5]. 

Smart textiles are commercially available [6] in different applications such as in sports [7], healthcare [8], vehicle industry [9], military [10], personal protection, and safety and space exploration [6]. Although smart textiles are used in all spheres of our lives, healthcare is the most remarkable market area, potentially enabling the development of new healthcare systems that can ensure significant cost reductions [11]. Due to the increasing complication of medical treatments on the one hand and the advancement of technology in the area, on the other hand, there is an emerging trend for a personalized healthcare system. Smart textiles have the lion’s share in this regard, where uses are equipped with wearable sensors to monitor their vital signs continuously and they give greater potential to users to take active control of their health as part of a preventive lifestyle which brings a reduction in healthcare cost by the early detection of health problems [5,12]. 

Devices that monitor physiological activities such as heart activity or electrocardiogram (ECG, also called EKG) [13], brain activity/electroencephalogram [14], muscle activity/electromyography [15], and other health indicators such as skin temperature [16], respiration [17], breathing [18], sweating rate [19], etc. have tremendous advantages in monitoring health. ECG is the process of recording the electrical activity of the heart, one of the most important physiological signals, which contains a treasure trove of information about the heart condition and heart-related diseases, such as arrhythmia, cardiac arrest, premature atrial contraction, premature ventricular contraction, congestive heart failure and coronary artery disease [20,21,22]. Recently, wearable ECG devices that enable us to continuously monitor heart activity are being developed as textile-based devices. As textiles are an indispensable part of our life, this is quite convenient for handling [23]. In the review paper by Pain et al. [11], a survey on textile electrodes for ECG monitoring and the different materials used to develop textile electrodes, factors that affect their performance in the signal acquisition were presented. 

This paper aimed to provide a scientific overview of textile-based electrodes for ECG monitoring, with the main emphasis on the different types of electrodes and recent advancements on ECG electrodes in general, and dry textile electrodes in particular. Developments in wearable health monitoring clothes and conductive textiles, especially for ECG monitoring, are also addressed.

## 2. Overview of the ECG Signal 

Heart-related problems, which are called cardiovascular diseases, are among the most prevalent causes of death worldwide, killing at least 20 million people every year, which covers more than 30% of total deaths [24,25]. What makes the situation worse is that the patients become aware of their situations only after they become victims and often after it becomes severe. Therefore, it is important to develop appropriate and affordable heart activity monitoring devices to know the status at any time and detect disorders at an early stage. ECG is one of the most widely used vital signal sensing and health monitoring methods which contains important information to diagnose cardiovascular diseases and examine their development [26]. ECG is a medical diagnosis activity for the heart, performed by placing two or more electrodes on the skin [20,27]. The heart pumps blood throughout the body, and during this time an electrical signal is generated. The heart contains upper and lower parts, the upper chambers are called atria, and the lower chambers ventricles; each part consists of a right and left chamber. During normal blood circulation, the right atrium receives deoxygenated blood returning from the body and the left atrium receives oxygenated blood from the lungs. Similarly, the left ventricle receives oxygenated blood from the left atrium and pumps it through the aorta and then out to the rest of the body, whereas the right ventricle receives blood from the right atrium and pumps it through the pulmonary arteries to the lungs, where it picks up oxygen and drops off carbon dioxide. Atria contract (depolarize) to pump blood to the ventricle and relax (repolarize) to receive blood and in a similar manner, the ventricles also contract during blood pumping and relax while receiving blood from the atria. During this heart activity, the blood circulates in our body and the heart generates electrical currents due to the polarization and depolarization of the atria and ventricles which can be measured on the skin using ECG [27,28]. The sinoatrial node, which is the natural pacemaker of the heart located at the right atrium, is the origin of the electrical activity of the cardiac system. Conducting pathways take the impulse to different parts of the heart to regulate heartbeats [29,30].

The full morphological structure of the ECG signal contains the P wave, T wave, and QRS complex as shown in Figure 1a, where the P wave represents the depolarization of the atrium, the QRS complex is generated by the depolarization of the ventricles, and the T wave results in the re-polarization, i.e., relaxation of the ventricles [31,32]. The quality of the ECG signal is essential for precise and accurate heart activity monitoring [24]. Any ECG device, whether clinical or portable, contains three essential components: the electrodes which are placed over the skin to capture the ionic currents generated from the heart and convert them into electrical current, the interconnections or wires taking out the acquired signal to the processing unit, and the processing unit which is used to filter unwanted signals and amplify the signal to help identify and interpret the signal [26,33,34]. Of these, the electrodes are the main components that affect the quality of the acquired signal. Conventionally, ECG is recorded by placing the electrodes on the skin, usually around the chest and on the limbs. The standard clinical ECG acquisition employs a 12-lead system that contains 10 electrodes with six electrodes fixed around the chest labeled V1 to V6 and four electrodes attached to the limbs [35]. Among the limb electrodes, three electrodes are used to generate three bipolar limb leads (Leads I, II, and III, which record the voltage difference between two limb electrodes). Lead I refer to the voltage between the left arm and right arm electrodes, Lead II refers to the voltage between the left leg and right arm electrodes, and Lead III refers to the voltage between the left leg and left arm electrodes. Unipolar limb leads, which are also called augmented leads, are derived from the same three electrodes used for the bipolar leads. Augmented vector left (aVL) lead is recorded by placing a positive electrode on the left arm and the negative pole is a combination of the right arm electrode and the left leg electrode, while the augmented vector right (aVR) lead uses a positive electrode on the right arm and a negative pole which is a combination of the left arm electrode and the left leg electrode. In the same way, the aVF lead uses a positive electrode on the left leg and the negative pole is a combination of the right arm electrode and the left arm electrode. The augmented leads can be computed based on the bipolar ones. Here, the fourth electrode is used to provide a ground reference, usually through active circuits. Figure 1b shows the standard electrode placement in the 12-lead system. Usually, ECG measurements using textile electrodes use only two or three electrodes [24], though in the case of two electrodes, a single bipolar limb lead can be obtained, while with three electrodes, Leads I, II and III can be obtained.

ECG diagnosis is usually performed at a hospital and only lasts for a few minutes, typically 10 s in a static position. As many heart problems demonstrate themselves infrequently or are triggered by a particular activity, the chance that they are observed during such a short test time is small and may lead to a false analysis of the heart condition of the patient by the physicians. Therefore, to properly monitor heart activity and diagnose possible abnormalities in advance, it is useful to have long-term continuous ECG recording [21,36]. Portable cardiac monitoring devices such as Holter monitors have been used to record a continuous ECG usually for 24–72 h [11]. However, these portable devices are inconvenient for the person who wants to record their activity because of the electrodes they employ and the wires. Lightweight wearable ECG devices that use miniaturized electronic components would be ideal for continuous monitoring. These could be valuable to undertake preventive action against heart disorders at an early stage prior to a serious condition occurring; therefore, wearable ECG devices could play a major role in minimizing the rate of mortality caused by heart-related diseases.

## 3. ECG Electrodes Results

Electrodes are sensors for electric phenomena in the body, which are attached to the skin and collect the electrical currents [37]. As these electrodes enter the vicinity of the skin, they form a skin–electrode interface and allow the exchange of electrons or ions. Skin moisture, i.e., sweat, is used as an electrolyte interface that will occur naturally after the placement of an electrode [27]. Based on the principle of current flow, electrodes are classified as non-polarizable and polarizable electrodes. Electrodes cannot be perfectly polarizable or non-polarizable; however, certain classes of electrodes can approximate these characteristics, as Ag/AgCl electrodes are examples of non-polarizable electrodes. In perfectly non-polarizable electrodes, there is an actual current between the skin and the electrodes. Such electrodes require electrolyte gel or electrically conductive skin interfacing materials containing dissolved electrolyte salt and oxidizing agent to improve their performance. Whereas in perfectly polarizable electrodes, there is no actual electrical current flow, instead, there is the only displacement of current due to charge accumulation around the skin–electrode interface, and the electrodes work based on the capacitance detecting technique between a conductive material and the skin [11,27,38]. Figure 2 shows a schematic representation of current flow in ideal non-polarizing and polarizing electrodes.

Disposable silver/silver chloride (Ag/AgCl) gelled electrodes are most commonly used for ECG signal measurements, both for clinical-based and portable devices [25,39,40]. These electrodes employ an electrolyte gel to reduce the skin–electrode interface impedance to have a good electrical contact between electrode and skin and generate a high-quality signal [20]. However, Ag/AgCl-gelled electrodes have several drawbacks. Especially during long-term signal monitoring, their disadvantage becomes more obvious. This is because the gel used in Ag/AgCl electrodes dehydrates gradually during usage, affecting the signal quality and causing skin irritation, redness, dermal inflammation, and other skin problems [25,41]. To guarantee signal stability during prolonged recordings, this gel needs to be regularly reapplied to obtain a good quality signal, as the signal quality deteriorates due to the drying out of the gel over an extended period [42]. Besides, excessive hair must be shaved and a gentle scrape of the skin is required to remove the outer layer of the skin (stratum corneum) which contains dead cells that have high resistance and lead to high skin–electrode contact impedance reducing signal amplitude. This important preparation of the skin to reduce the thickness of the outer layer is required to improve the skin–electrode contact and acquire good signal quality. However, these extensive skin preparations and the drying out of the gel over time requiring continuous replacement of the gel are time consuming and uncomfortable for both patient and physician, especially for the long-term continuous recording of bio-signals [34,43,44]

The drawbacks of the conventional Ag/AgCl gelled electrode for long-term monitoring and the growing interest in home care bio-signal ECG sensing devices led to the development of dry electrodes containing no liquids or gels [25,42,45]. Dry electrodes are suited for long-term and continuous ECG monitoring without the need for applying gels at intervals [46]. The earlier forms of dry electrodes are metal disks made from rigid metals such as stainless steel. However, such rigid disk dry electrodes have high contact impedance [47] and show motion artifacts in the ECG signal caused by variations in the contact area during body movements. Furthermore, they lack flexibility which hinders their adaptation to the changing body topography and conformal contact with skin [22,48]. The aforementioned limitations make dry metal disk electrodes unsuitable for wearable applications. The contact impedance of metal disk electrodes can be reduced by using micro-needle array structure electrodes [47,49,50]. Electrodes with microscale needles (spikes) on the surface penetrate the outer part of the skin. In order to function well on hairy skin, contact electrodes are often equipped with macroscale pins, since the hair can be positioned in the space between the pins and hence results in skin–electrode contact [50]. Albulbul studied the skin–electrode impedance properties of wet Ag/AgCl electrodes and compared them with orbital electrodes with pins (spikes) of 150 µm height and flat stainless-steel electrodes [47]. He reported that the Ag/AgCl electrodes showed the lowest skin–electrode impedance followed by orbital electrodes. Orbital electrodes provided lower impedance than the stainless-steel electrodes due to the presence of pins or spikes, which is essential to acquire high bio-signal quality. However, this approach can cause severe damage to the skin due to the penetration of the microneedles and the needles will easily break during use, leaving needles behind in the skin [20,50]. Figure 3 presents standard Ag/AgCl electrodes and metal-based spiked and flat electrodes with their internal and external side. Although several dry electrodes have been developed using metallic materials, they have been limited in their practical use due to high skin–electrode impedance, poor biocompatibility, and variations in the contact area during motion. Dry electrodes made from textiles could be good alternatives for wearable physiological signal monitoring because of their air and water permeability which will prevent or reduce skin irritation and other skin problems arising with gelled and rigid metal electrodes [11,41].

## 4. Textile-Based ECG Electrodes 

Wearable electronic smart textiles can be developed by using the textile fabric itself as a sensor or embedding the sensor in textile clothes. The integration of flexible ECG sensors with everyday textiles will be convenient for handling and cost-saving purposes [11,20,51]. Instead of attaching a separate electrode like a disposable Ag/AgCl electrode, making the textile itself a sensing electrode is more interesting for monitoring the health and wellbeing of individuals demanding long-term heart monitoring. Textile-based sensors could contribute possibilities for providing more affordable, accessible, and easy-to-wear measuring devices, thus giving a greater potential to users to take active control of their health as part of a preventive lifestyle which brings a reduction in healthcare cost by the early detection of health problems. However, conventional textile products that are found in everyday garments are intrinsically electrically non-conductive and hence cannot be directly used for bio-sensing applications. Sensing or data transmission via the textile material requires the textile to be electrically conductive. Textiles can be made electrically conductive by integrating metals, carbon materials, or conductive polymers into the textile structure through several techniques at different stages (fibers, yarns, or fabrics) [48,52,53,54,55,56,57,58]. 

Wearable textile electrodes for continuous health monitoring products, as part of standard clothing, are needed to satisfy several requirements. The most important requirement is that they should have adequate electrical conductivity [11,59]. Sufficient electrical conductivity is necessary to detect even small amplitudes in the electrophysiological signals of the heart. High electrical conductivity results in lower skin–electrode impedance which is very essential during ECG measurements to acquire high signal quality [60]. Additionally, as part of standard clothes, textile electrodes should have a good visual appearance taking into account fashion aspects, and at the same time be comfortable for the wearer [61]. Furthermore, the conductive textiles should allow standard maintenance such as washing and ironing. In addition, the electrodes should be easy to wear and use, and should be as lightweight as possible, though should not hinder the users’ movement and daily activity [62]. Apart from the electrodes, the signal recording also needs interconnections and data processing and possibly an antenna, which should also be integrated (completely or partly) into the garment. After data processing, the result may then in turn be used to display information on the health status of the wearer, either to the user or to the concerned people such as their physician. All these parameters of wearable sensors make designing such products very challenging as many conflicting requirements must be considered during product development.

Depending on the coupling between the electrode and the skin, textile-based dry electrodes can be categorized into two types: contact and non-contact electrodes. In contact electrodes, the direct physical coupling is established between the skin and the electrode [39,63]. Electrodes are required to have continuous conformal contact with the skin of the wearer to allow consistent signal detection and to minimize artifacts and noise, i.e., unwanted signals [60]. When electrodes are directly attached to the skin, they should be biocompatible not to cause any negative impact on the skin of the user, whereas in non-contact electrodes, there is no physical contact to the skin and the electrodes are rather separated from the skin by a dielectric material or air [64]. In a non-contact system, the electrodes should be kept at a fixed distance from the skin for optimal operation. These sensors function by the principle of sensing the electric field created by the displacement currents in the body through the coupling of charges between the patient’s skin and the electrode and are called capacitively coupled electrodes [21,65]. Non-contact electrodes are especially important when contact electrodes damage the skin like in newborns. Most electrodes used are contact electrodes and provide better signal quality. These textile-based electrodes can be developed by integrating yarns or wires to the textile structure or applying conducting compounds onto the textile fabric surface. In the following section, the methods to make textile electrodes for ECG applications have been covered in detail.

### 4.1. Metal Integrated Textile Electrodes

The earlier forms of electrically conductive textiles are made by integrating metal yarns into textiles. Metal yarns are different from metal wires, as they consist of metal fibers or filaments that are processed as standard textile fibers (cotton or polyester) to create a yarn. They were first developed to discharge static electricity. The metal fibers are produced either through a bundle-drawing process or shaved off the edge of thin metal sheeting, leading to very thin metal filaments (diameters ranging from 1 to 80 micron). Metal fabric electrodes can then be developed by integrating metal yarns made up of these metal fibers such as stainless steel during the manufacturing stage (weaving, knitting, or nonwoven). 

Recently, a lot of metal integrated textile-based ECG non-contact electrodes have been developed by many investigators [66,67,68]. The metal threads in this can be silver, copper, or stainless steel, but also other metal wires are employed to develop conductive textiles. Li et al. made a textile-based non-contact ECG acquisition system based on capacitive coupling textile electrodes from conductive textiles with stainless steel [66]. Several works [67,68] have been reported on capacitive coupling ECG monitoring from different flexible dry textile electrodes. These non-contact electrodes can be integrated into clothing [69], a chair [70], a wheelchair, a hospital bed, and a stretcher [65], or other seats. Some examples of non-contact textile ECG sensors are presented in Table 1.

The major challenges in designing these non-contact electrodes lie in dielectric change due to their indirect contact and the capacitive mismatch caused by motion artifacts due to subject movement, respiration, and electrode impedance changes which cause low signal-to-noise ratio [73]. Overall non-contact electrodes are very sensitive to any motion of the electrode with respect to the body as the capacitance changes dramatically, and the insulating material relied on between the skin and the electrode hinders current transfer [42]. To improve signal quality, additional active circuits with buffers are used together with the electrodes to filter the ECG signal resulting from such capacitive electrodes [45,69,74]. Even though this improves signal quality, using additional hardware requires a larger amount of energy as well as additional wires which need further investigation to make a convenient wearable system [60]. An elastic belt that presses the electrode and the dielectric barrier to the skin is the best option to obtain a good quality signal, as such pressure reduces the variance leading to noise [67]. Such a system is hence based on the non-contact principle but uses pressure to improve quality. Some examples of non-contact metal integrated textile electrodes are shown in Figure 4a–c embedded in different systems. In Figure 4d, comparisons of the ECG signals collected using standard Ag/AgCl and non-contact textile electrodes are given, and it was reported that their mean correlation was 0.96 [71].

The currently available contact-based wearable metal integrated textile electrodes for ECG monitoring systems use several types of electrodes. Rienzo et al. used a textile fabric vest composed of textile sensors made of two woven electrodes made up of conductive fiber placed around the thorax to collect ECG signals [76]. The contact between the electrode and body skin was guaranteed by integrating the electrodes in an elastic garment, i.e., the vest made up of cotton and lycra without using any gel. The system was used to monitor cardiac rhythm and arrhythmic events in cardiac patients during static and dynamic conditions and the signal quality was identical when compared to that of the traditional ECG devices. It was interesting that it even showed slightly better results when the recording was done during physical exercise due to better contact with the body by the elastic vest [77]. The holding pressure of the electrodes by the elastic vest or band affects the skin–electrode impedance, which in turn directly affects the collected ECG signal quality. It has been reported that increasing the applied pressure (tightness) causes a decrease in skin–electrode impedance and motion artifacts [78,79]. Ankhili et al. [80] have developed an embroidered textile electrode from two types of silver-plated polyamide conductive threads and assessed their ECG detection performance up to 50 washing cycles. 

The effect of the size of electrodes on signal quality has been studied by several works [81,82,83]. The skin–electrode impedance decreases as the electrode size increases due to an increase in the contact area [71,83], which minimizes the baseline effect while collecting ECG signals [81,83]. Joutsen et al. [82] studied four different sizes of stainless-steel conductive textile electrodes (with 40, 20, 10, and 5 mm diameters) embedded in a textile elastic strap to hold the electrode in place for ECG and heart rate monitoring. It was shown that 40 and 20 mm diameter stainless steel electrodes gave better results and are more suitable for textile integrated ECG and heart rate monitoring, indicating that a larger size electrode improves the signal.

Even though metal threads provide good conductivity, which is very essential to be used as electrodes since they can easily detect the electrical activity of the heart over time, they still have limitations for use in wearable sensors. This is due to the fact that metal threads are heavier in weight compared to standard textile fibers, have limited flexibility affecting the fabric softness, are abrasive and unpleasant when in contact with the skin resulting in skin tingling, and cannot easily meet the requirement of being washable as some meal threads easily break during washing [84,85]. The metal must also be biocompatible, making most metals unsuitable. 

Another type of metal-based conductive textiles for biopotential sensors can be produced by coating the textile products with conductive metal nanoparticle inks such as silver and stainless-steel inks, rather than weaving or knitting the metal yarn during textile manufacturing. Vojtech et al. [59] developed textile-based ECG monitoring, in which the electrodes were made up of knitted fabric containing polyester coated with silver nanoparticles integrated into a T-shirt, which easily allowed standard textile maintenance such as washing and ironing. The presence of silver nanoparticles provides corrosion resistance to the electrodes, antibacterial and antiallergic properties, as well as mechanical and electrical stability when exposed to sweat. They reported that the ECG signal quality recorded while the individual was lying on the bed in the supine position showed a clear signal morphology with well recognizable P waves, QRS complexes, and T waves. 

Dry textile-based electrodes are sensitive to motion artifacts because of the movement of electrodes in relation to the skin while the person is in a dynamic condition. Tong et al. [86] studied the sensitivity of the textile-based electrodes to skin–textile contact, textile placement, user activity, and muscle activity. ECG recording was performed while varying the applied compressive force from 2 N up to 10 N to vary skin–textile contact, placing the electrodes in different positions on the arm, considering at rest and walking conditions as well as when holding a gripper to evaluate the effect of muscle contraction on the ECG signal quality. Results showed that better signal quality is obtained when the compressive force is ≥6 N and while the user is standing. In this case, the acquired signals have high accuracy on heart rate measurement. An ECG electrode developed through silver paste screen printing on cotton and polyester fabric was reported in [87], and it was reported that 15 mmHg pressure is required to avoid excessive motion artifacts. However, too high tightness might cause discomfort and even affect blood circulation. 

Metal-coated conductive textile materials are attractive for biopotential monitoring due to their high conductivity; however, they have certain limitations. Some metals are prone to corrosion, especially under conditions such as humidity, metals such as stainless steel have a high density which affects the fabric weight and softness, and metallic-coated fabrics have poor abrasion resistance [84]. Some examples of metal-coated textile-based ECG electrodes are shown in Figure 5a–c, Figure 5d presents a comparison of ECG signals collected using Ag/AgCl and silver-coated textile electrodes, and results revealed that major peaks are visible in both signals.

### 4.2. Carbon-Coated Textile Electrodes

As the interest in conductive textiles increases, alternative methods are being investigated to produce suitable products for wearable smart textiles. Carbon materials have been used in the development of electrically conductive polymer composites and textiles. Carbon materials have outstanding potential for producing conductive textiles due to low cost, corrosion resistance, flexibility, excellent electrical properties, and high aspect ratio (length-to-width ratio) [89,90]. Carbon-based materials such as carbon fibers, carbon nanotubes (CNT), and chemically modified graphite and graphene (GN) have been used in the development of conductive textiles for biopotential monitoring [90]. Yapici and Alkhidir [39] reported graphene-clad conductive textile sensing electrodes which are washable. A wearable smart medical garment for ECG monitoring was developed by stitching the electrodes on an elastic wristband and neckband. The test results revealed that these signal qualities collected by the dry electrodes were comparable with conventional Ag/AgCl electrodes. Lee and Yun [20] proposed a wearable ECG monitoring garment that employed patch electrodes made of conductive carbon-based paste (p-electrode) that is directly applied to the skin. ECG monitoring feasibility was evaluated by using a tight-fitting elastic sport shirt, where small silver discs created the electrical contact of the p-electrodes and wires [20]. The contact impedance between the patch electrode and the skin was low (70.0 kΩ), whereas the contact impedance of Ag/AgCl electrodes was 118.7 kΩ. This was due to the carbon paste which covered the skin and provided a conformal contact with the silver discs in the t-shirt, which helps to provide excellent ECG signal quality at different conditions comparable with conventional gelled electrodes. Figure 6a,b present textile ECG electrodes and Figure 6c,d show a comparison of ECG signals collected using textile and Ag/AgCl electrodes from [40,54]. In [39], the cross-correlation of the entire waveform in Figure 6c was 88%, while in [63], the cross-correlation in Figure 6d showed an almost perfect overlap of signals.

The electrical conductivity of carbon materials is comparatively low when compared to that of metals and therefore increasing the amount of carbon material decreases the resistance for textile materials. However, this hurts the mechanical property of the textile [89,90]. Additionally, studies on biocompatibility also show that there are also toxicological concerns regarding CNTs [91].

### 4.3. Conductive Polymer-Coated Textile Electrodes

Recently conductive polymers have received much attention as they allow for the creation of lightweight and flexible conductive materials, such as textiles coated with a conductive polymer. These materials, owing to their flexibility, durability, ease of manufacturing, and application, are considered promising for wearable health care applications. Moreover, conductive polymer-based conductive textiles are expected to allow for the creation of more comfortable textile electrodes for ECG applications. Recently, a lot of work on conductive polymer-based textile electrodes has been reported, owing to their promising electrical conductivity, ease of use, and low process cost [92,93]. They do not affect the flexibility of the inherent textile while having better abrasion resistance than metal coatings and reasonable environmental stability [58,94]. Conductive polymers such as polypyrrole (PPy) [95], polyaniline (PANI) [22], and poly-3,4-ethylenedioxythiophene doped with poly (styrene sulfonate) (PEDOT:PSS) [23] have attracted the interests of researchers to develop conductive textiles. 

Conductive polymers can be applied to the textile, either by directly polymerizing the polymer on the textile from a monomer or by coating an already polymerized polymer on the textile. During polymerization, the monomers are polymerized on the textile substrate by in situ vapor phase or electrochemical polymerization through the application of an appropriate oxidant [59]. By applying conductive polymers directly onto textiles, a thin, homogeneous coating can be achieved on the surface of the textile. This enables a combination of a wide range of structural and mechanical properties of textiles with the electrical properties of the conducting polymers, providing infinite possibilities in the design. The thickness of a conducting polymer film formed during chemical synthesis, which affects both the resistance and flexibility of the electrode, is dependent on the synthesis time and reactant concentrations and it is usually in the submicron range [24,58]. Figure 7a,b present textile ECG electrodes made with PEDOT:PSS.

The most common method of preparing conductive textile electrodes using conductive polymers is by a coating, which can be either dip-coating or printing. This method is quite promising as it is compatible with the current commercial textile processing and will be a cost-effective method of treatment. Several works reported the fabrication of textile electrodes for ECG recording by the screen printing of PEDOT:PSS conductive polymer over the textile substrate [96,97,98,99,100]. This gave good ECG signal quality even though they had high skin–electrode contact impedance due to low contact pressure and unstable contact. Ankhili et al. developed PEDOT:PSS-coated polyamide textile-based electrodes using two different PEDOT:PSS solutions and evaluated the performance of signal quality after 50 washing cycles (Figure 7c,d) [92]. The results revealed that, even though there is a decrease in signal quality after repetitive washing, the results remain acceptable for ambulatory purposes. Wattal et al. reported on an ECG monitoring t-shirt with two electrodes/connectors made from conductive Velcro [95]. A pyrrole polymerization process was used to make the Velcro electrically conductive while also acting as a fastener to improve skin–electrode contact. The textile electrode/connector showed a stable resistance of 76 Ω over 300 connection cycles (attaching and detaching each connector), however, resistance increased from 71.5 Ω to 347.5 Ω after 10 machine washing cycles. 

The effect of body motion on ECG signal quality and long-term stability has been studied by many investigators [26,81,96,101]. Takamatsu et al. [23] used electrodes that were made up of knitted fabric coated with PEDOT:PSS and ionic liquid (IL) gel and compared with signals collected from standard Ag/AgCl electrodes. The results revealed that ECG signals recorded using textile and Ag/AgCl electrodes at sitting position are comparable as shown in Figure 8a, with 16.3 dB (±0.1 dB) SNR for both electrodes. However, ECG signals measured during movement (Figure 8b) show much lower quality though the textile electrodes are still usable. Specifically, the textile electrodes (in blue) show high signal content with a defined R peak due to the presence of IL gel that promotes better skin–electrode contact during body motion. Similarly, signals recorded from the textile electrodes show (Figure 8c) better accuracy of heartbeat detection during different types of activities. Long-term signal stability study results (Figure 8d) show that signals are consistent and there was no skin effect even after 3 days of use, while the signal recorded after one-month storage of the electrode still provided clear PQRST waveforms. Similarly, in [96], textile ECG electrodes developed via the dip coating of conventional fabric using PEDOT:PSS were reported. The PEDOT:PSS-coated fabric sewed to nonconductive layered of foam and polyester. The results revealed that signals collected at static and dynamic conditions using textile electrodes show comparable signal quality with signals collected using standard Ag/AgCl electrodes.

The main limitations for the use of conducting polymers in commercial applications are the fact that some polymers are unstable in air, have relatively low electrical conductivity, have poor mechanical properties, as well as the brittleness of films formed by the chemical or electrochemical synthesis methods [58], all of which require further research to allow these conductive polymers to be used in real products. Table 2 shows a non-exhaustive list of contact-based textile ECG sensors with the type of conductive materials used and electrode placement.

## 5. Textile Electrodes in Veterinary ECG Monitoring

ECG monitoring is an important health diagnostic tool in veterinary medicine [112]. Long-term ECG monitoring is very important in veterinary clinical practice to diagnose the health condition of animals. Studies have been performed on sheep [113], dogs [114], buffaloes [115], rabbit [33] and horses [116]. Conventionally, crocodile clamps electrodes together with adhesive gel are used for ECG measurements on animals. As the clamps pinched to the skin of the animal can cause pain and the application of gel could bring skin reaction, these electrodes are considered inconvenient [116]. Alternatively, self-adhesive wet electrodes were used, however, as the skin of animals is typically covered with long hair, their drawbacks become more obvious as such electrodes are not suitable for long-term ECG monitoring. The demand for long-term ECG monitoring in veterinary medicine led to the development of dry textile electrodes that do not require gel electrolytes. As textiles are inherently lightweight and flexible, textile electrodes are convenient for long-term ECG monitoring both in human and veterinary clinical practice [117]. 

Studies have been conducted on the development of textile electrodes for veterinary electrocardiography (ECG) monitoring. Guidi et al. [118] compared the ECG signal acquisition performance of textile electrodes and Ag/AgCl electrodes in a horse. The textile electrodes were fixed in place using an elastic belt around the chest without any skin preparation, and sponges were placed between the elastic belt and the electrodes to provide better skin–electrode contact. The standard Ag/AgCl electrodes on the other hand were attached to the shaved skin with self-adhesive pads. They reported that ECG signals collected with textile electrodes show lower motion artifacts than Ag/AgCl electrodes. Similarly, Felici et al. [119] presented the ECG monitoring of horses during exercise on a treadmill using textile electrodes, and the results were compared with signals from standard Ag/AgCl electrodes. The textile electrodes provided comparable signal quality during treadmill exercise. However, they did not study the effect of washing, bending and multiple uses of the textile electrodes on ECG acquisition and whether the textile electrodes maintain their textile texture and property. Similarly, the type of textile material and structure was not mentioned.

Recorded ECG signal quality is affected by motion artifacts and remains one of the major problems which arise due to the absence of uniform skin–electrode contact, the detection of muscle movement, higher contact impedance, and unwanted wire movement during acquisition [113,120]. Motion artifacts can be minimized via proper skin preparation that could help to reduce contact impedance and by using modified electrodes. Even then, there is a need for algorithms that detect and remove artifacts from the acquired signals, especially for animals, as they cannot be told to stand still [120]. Lanata et al. [120] presented an algorithm called stationary wavelet movement artifact reduction to detect and reduce movement artifacts (MAs) in ECG signals collected from the horse through a wearable system that employs textile electrodes. They reported that after the application of the proposed algorithm, the results achieve a 40% reduction in MA which was higher than obtained with a normalized least mean square adaptive filter technique. A wearable system for ECG monitoring in a horse was published by Guidi et al. [117] and they estimate human–horse interaction by analyzing heart rate variability using a dynamic time warping algorithm which also compared their performance with standard Ag/AgCl electrodes in terms of MA percentage.

## 6. Design Aspects

As an alternative to wet electrodes, dry textile electrodes that do not require gel electrolytes are suitable for wearable long-term biosensing to monitor cardiovascular diseases. Textile electrodes could be used in a wide variety of applications and have a larger contact area compared to other dry rigid non-textile electrodes, but when embedded in standard clothes, they often do not provide complete conformal contact with skin. Conductive textiles are created by coating a conductive component on the textile surface or by integrating metal fibers during the manufacturing stage, but because of this, the flexibility of the fabric is negatively impacted, and the fabric is no longer able to fit conformably with the curvilinear surface of the skin. For this reason, textile electrodes are usually integrated inside tight fit garments during the manufacturing stage [121] or patches of textile electrodes are embedded inside elastic bands or tight fit shirts [116] to keep the electrodes in their position and to ensure the proper connection to the skin of the user [78]. However, nonetheless, the electrodes create minimal movement over the skin surface, which causes variation in skin–electrode impedance and negatively affects the signal quality by generating noise interferences [11]. To improve skin–electrode contact, prevent displacement of the electrodes on the skin surface during body motion, and ensure uniform holding pressure, some researchers prepared textile electrodes in a sandwiched structure by sewing pieces of conductive fabric with a foam layer and an outer layer of non-conductive fabric [101,122], or simply sew the conductive fabric to a non-conductive synthetic leather [96]. Moreover, slightly moisturizing the skin or the surface of the textile electrode with a spray of tap water (saline solution) helped to lower skin–electrode contact impedance and as a result, improved signal quality [97,121], which shows the skin moisture (sweat) would also help to improve signal quality. Textile is especially suited for this due to the capillary suction that can retain the water. Weder et al., developed a breast belt with an embroidered textile electrode together with a small water reservoir (Figure 9f) to keep the skin–electrode interface humid and to obtain good signal quality [103]. However, keeping the skin moist for a long period may cause discomfort to the users. On the other hand, in [23], an ionic liquid gel is used to enhance continuous skin–electrode contact and to improve acquired signals using textile electrodes under dynamic conditions. Figure 9 presents some design examples of wearable textile ECG monitoring systems.

Research outputs prove that textile electrodes are full of motion artifacts because of an unstable contact of the electrodes on the skin leading to high skin–electrode contact impedance [124,125,126]. The skin–electrode contact impedance depends on contact pressure, electrode placement, user activity and muscle activity in addition to the main textile electrode impedance [123]. Similarly, currently available wearable health monitoring devices employ cables to connect electrodes to a data processing unit which hinders user activity and leads to discomfort [40]. Due to this, they are not in routine clinical use yet [127]. Designing a wearable ECG monitoring system that guarantees a permanent skin–electrode contact and allows high ECG signal quality, which would be important to develop a true textile wearable system. Additionally, the washability and the dimensional and environmental stability of the electrodes are key issues for long-term use that should be considered.

Generally, previous research in the area of textile-based electrodes has overcome several problems associated with wearable biopotential monitoring devices, however, the comfort of the user and the ECG signal quality is not at a satisfactory level [59]. It is possible to overcome some of the drawbacks of textile electrodes through optimal design, such as higher contact pressure, moisturizing, increasing electrode size, and others. However, many of these changes lead to other problems, mostly concerning the comfort of the user.

## 7. Future Research and Directions

The demand for wearable health monitoring electronic textiles is promptly increasing due to their flexibility, lightweight, and washing advantages over standard electrodes. The use of textile-based electrodes to monitor physiological activities also avoids the use of gel to reduce skin-to-electrode impedance. Therefore, such electrodes would be an ideal replacement for the gel-based wet electrodes and are promising electrodes for wearable applications for long-term monitoring.

Since the introduction of textile-based ECG electrodes in the 1990s, the research in finding better electrodes has drastically increased with much promising improvement in artifact minimization, impedance lowering, SNR minimization, and signal quality improvements. For instance, searching with “ECG + textile” results in 316 Web of Science indexed articles and conference proceedings that have been published in 30 different refereed journals in the last three decades (1990–2020). Moreover, the publication rate is increasing as shown in Figure 10. This shows that many experts are spending their time in the lab looking for improved textile electrodes. This pace of research should eventually result in finding working, washable and textile electrodes for long-term monitoring.

The rapidly increasing demand for e-textiles for physiological monitoring resulted in new conductive materials investigations. New state-of-the-art integration techniques of electronic components in the textile structure could also evolve as a worthy solution in the near future. For instance, the use of 4D printing and investigating 4D conductive materials that are able to change their size and dimension under external controlled stimuli could play a role in lowering skin-to-electrode impedance and improving fit and design concepts.

Smart wearable textiles are the result of interdisciplinary research, which connects concepts and expertise from textile engineering, computer engineering, electronics, material science, medicine, and more [5]. The integration of expertise and the high demand of markets for smart textiles leads to continuous improvements in the technology, supported by growth in research. The development of a reliable textile-based ECG electrode requires a collaboration of experts from textile, material, medical and electrical engineering.

## 8. Conclusions

With increasing health care costs and advancements in technology, there is an emerging trend for personalized health care. Wearable health care devices enable us to monitor different electrical physiological activities. Among the various bio-signals, ECG signals are one of the most important because they provide information about the heart condition and cardiovascular diseases which is one of the most prevalent causes of death worldwide. For the early detection and diagnosis of cardiovascular diseases, the continuous and long-term measurement and monitoring of the ECG signal is very important. The commercially used Ag/AgCl gelled electrodes are not suitable for long-term wearable health monitoring systems, due to dehydration of the conductive wet gel over time which causes signal quality deterioration and discomfort for the user, and because they are directly attached to the skin with a connecting wire, often leading to discomfort at the electrode location over time. Dry electrodes are considered a good alternative to those gelled electrodes, especially for long-term use. Dry electrodes made of rigid metals and conductive textile products have been reported by many investigators. 

Textile electrodes for ECG monitoring can be developed by integrating metal yarns into the textile or coating with metal nanoparticles, coating with carbon materials, and coating with conductive polymers. As textile clothing is one of the most frequently used materials in our daily life, they are ideal for wearable health monitoring systems. Wearable textile electrodes for continuous health monitoring products need to satisfy requirements such as high conductivity, aesthetics, and comfort, conformal skin–electrode contact, and biocompatibility. Although previous research on the area of textile-based electrodes overcame several problems associated with wearable biopotential monitoring devices, the comfort of the monitored person and ECG signal quality is still not at a satisfactory level. Mostly, textile electrodes are prone to motion artifacts due to higher contact impedance and the absence of conformal contact, which is why they have not yet achieved acceptance for clinical standard use. The current review showed that there are still a lot of challenges to be resolved for textile electrodes and wearable smart textiles to become more applicable in real-life situations and also become accepted by patients and other users as a reliable, multifunctional, easy-to-use, and minimally obtrusive technology that can increase their quality of life.

## Figures and Tables

**Figure 1 sensors-21-04174-f001:**
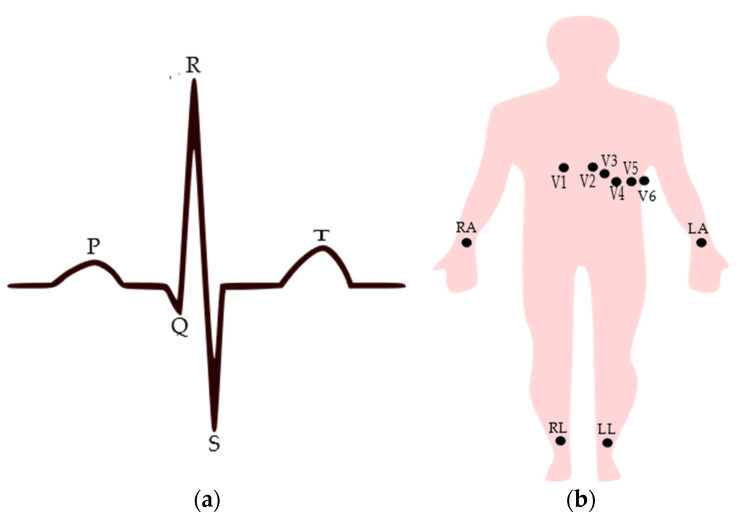
(**a**) Typical normal ECG signal with major components from Lead I configuration; (**b**) electrode placement for standard clinical ECG measurement, where V is the voltage, RA is the right arm, LA is the left arm, RL is the right leg, and LL is the left leg electrodes.

**Figure 2 sensors-21-04174-f002:**
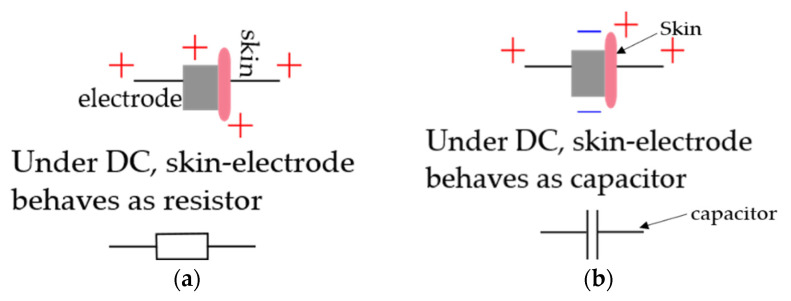
Schematic representation of: (**a**) ideal non-polarizing electrode; (**b**) ideal polarizing electrode.

**Figure 3 sensors-21-04174-f003:**
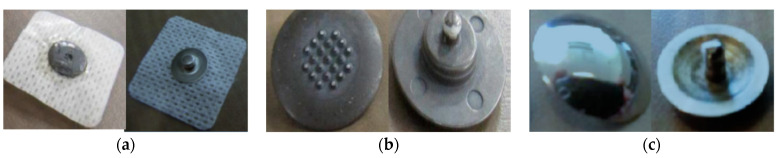
ECG electrodes with skin side (left) and snap side (right): (**a**) Ag/AgCl; (**b**) orbital with 150 µm pins; and (**c**) stainless steel electrode. Adopted from [47].

**Figure 4 sensors-21-04174-f004:**
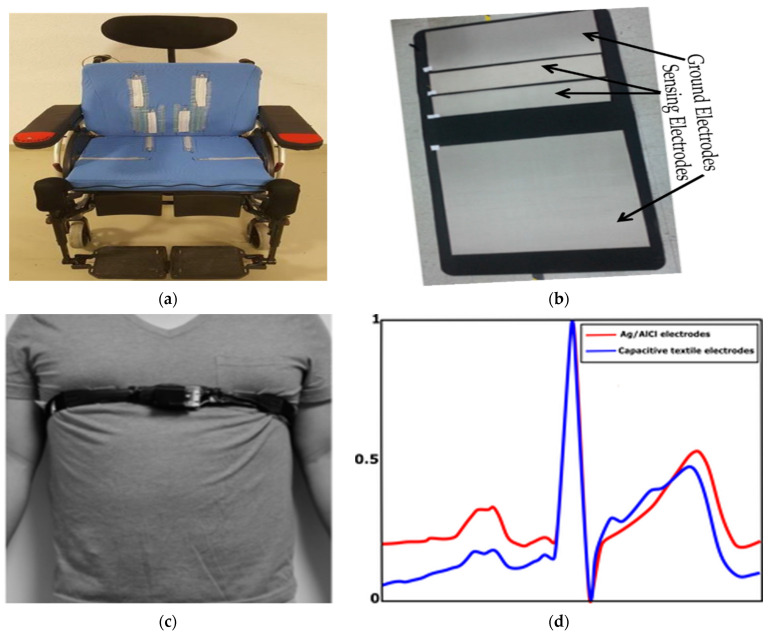
Non-contact metal integrated textile electrodes and typical results: (**a**) textile electrodes integrated into a wheelchair [75]; (**b**) smart mattress with textile electrodes, adopted from [71]; (**c**) a person wearing a strap with an ECG sensor (a foam coated with Ni/Cu) around the lower chest over her t-shirt, adapted from [67]; (**d**) normalized ECG waveforms measured from Ag/AgCl electrodes (red line) and capacitive textile electrodes (blue line), adapted from [71].

**Figure 5 sensors-21-04174-f005:**
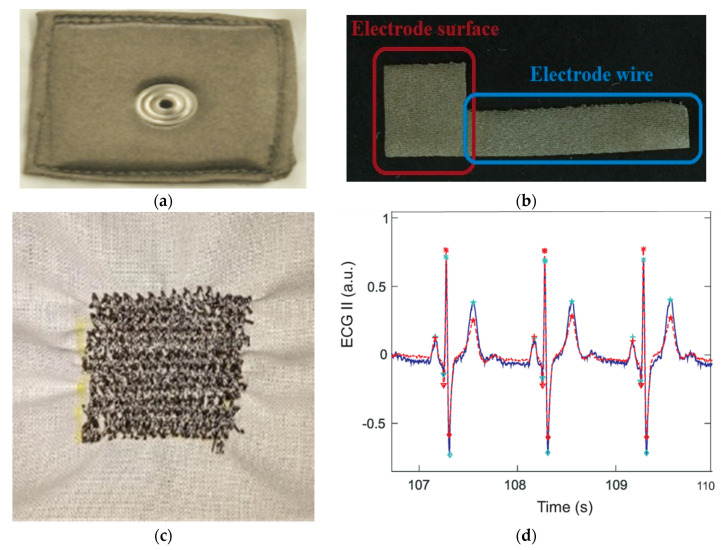
Examples of metal-coated textile-based ECG electrodes and resulting signal; (**a**) interior side of the silver-coated textile electrode, adopted from [74]; (**b**) structure of a textile electrode and connection track, adopted from [81]; (**c**) textile electrode developed by sewing silver thread onto fabric, adopted from [88]; and (**d**) comparison of ECG signals from a T-shirt with silver-plated active textile electrodes (blue) bandpass filtered (2–20 Hz) and reference Ag/AgCl electrode (red), adapted from [74].

**Figure 6 sensors-21-04174-f006:**
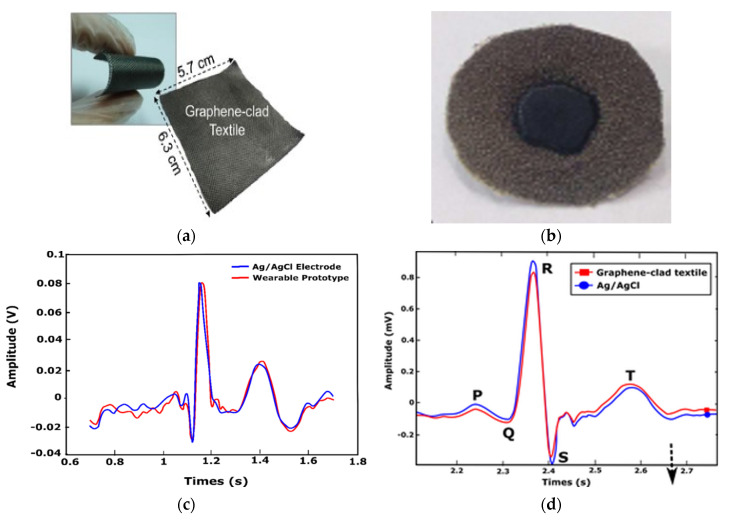
Examples of graphene-coated textile ECG electrodes and resulting signals: (**a**) photograph of a sample grapheme-clad nylon fabric for ECG electrode, adopted from [63]; (**b**) graphene-coated polyester fiber electrode, adopted from [25]; (**c**) ECG signals recorded from conventional Ag/AgCl electrodes and the graphene-clad textile with wristband, adopted from [39]; and (**d**) filtered ECG signals from Ag/AgCl and graphene-clad textile electrodes, adopted from [63].

**Figure 7 sensors-21-04174-f007:**
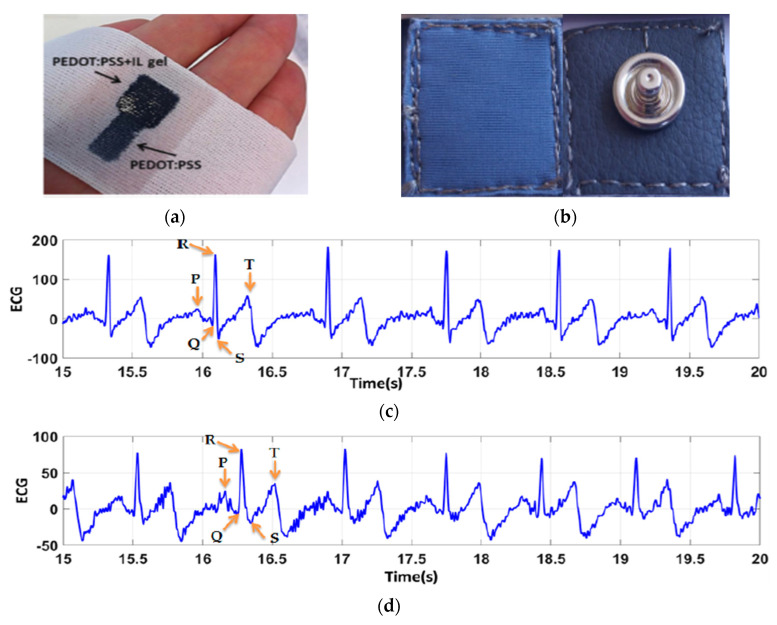
Conductive polymer-coated textile electrodes: (**a**) PEDOT:PSS and ionic liquid gel-coated polyester textile electrode, adopted from [23]; (**b**) PEDOT:PSS-coated textile electrode, adopted from [96]; (**c**) ECG signal collected from PEDOT:PSS-coated textile electrode before washing; and (**d**) after 50 washing cycle, adopted from [92]. The ECG signals collected using textile electrodes show 25.5 dB SNR before washing and 10.3 dB SNR after 50 washing cycles.

**Figure 8 sensors-21-04174-f008:**
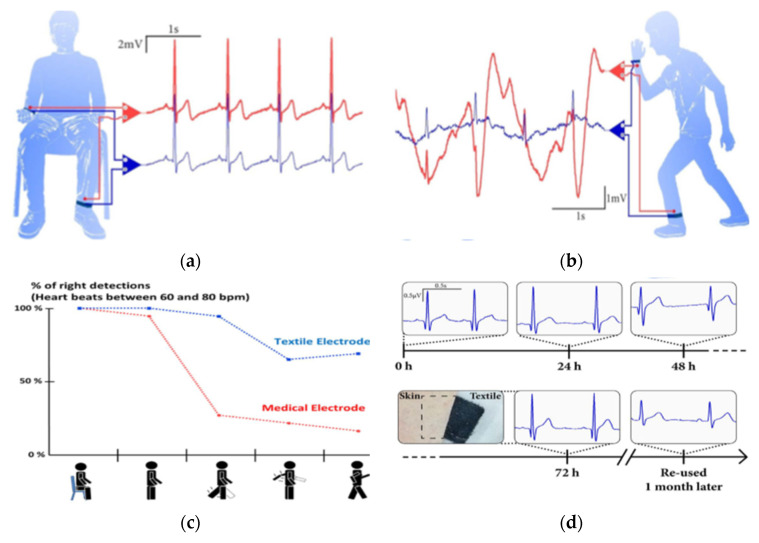
ECG recordings performed with the PEDOT:PSS/IL textile electrode (in blue), and Ag/AgCl electrode (in red): (**a**) from volunteers sitting at rest; (**b**) during movement; (**c**) percentage of the accuracy of heartbeat detection during different types of activity (seating, standing up, leg moving, arm moving, walking) with textile and Ag/AgCl electrodes; and (**d**) ECG signal evolutions obtained with textile electrodes in permanent contact with the skin over three days. The inset shows a picture of the skin under the electrode after 72 h. The last ECG signals were obtained from re-used textile electrodes stored in ambient air for one month, adopted from [23].

**Figure 9 sensors-21-04174-f009:**
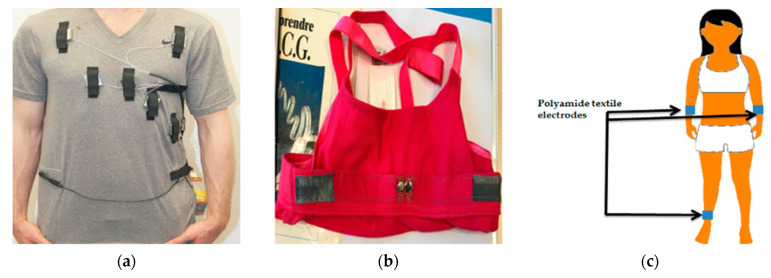
Different wearable textile ECG monitoring systems: (**a**) ECG T-shirt with active electrodes and connectors, adapted from [74]; (**b**) PEDOT:PSS-coated polyamide electrodes sewn into bras, adopted from [24]; (**c**) electrode placement for ECG measurement where plastic clamps were used to fix the electrodes onto the wrist, adopted from [92]; (**d**) ECG sensing wristband with printed and flexible electrodes, adapted from [91]; (**e**) wearable chest belt with silver-coated nylon woven electrodes and Bluetooth module, adopted from [123]; (**f**) ECG belt with wetting pad (above) and the embroidered electrodes (below), adopted from [103].

**Figure 10 sensors-21-04174-f010:**
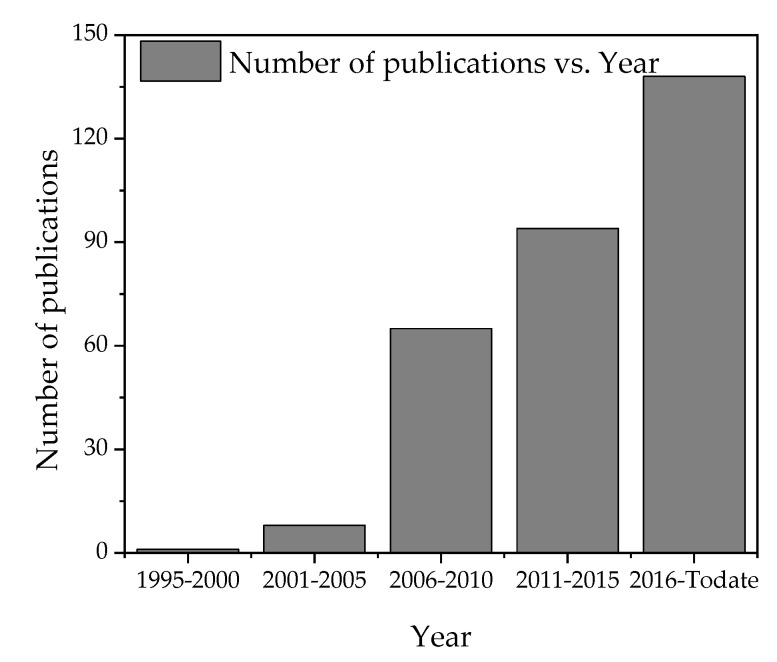
The progress of reports on textile-based ECG electrodes according to Web of Science core collections.

**Table 1 sensors-21-04174-t001:** Non-contact textile ECG sensors.

Integrated in	Materials for Electrodes	Type of Dielectric Material	Location of Measurement	Reference
Stretcher, hospital bed, and wheelchair	Silver gel-printed textile	Cotton T-shirt, sweater, and trousers	Backside	[65]
T-shirt	Polyamide covered with silver yarn and silver-coated by silver	T-shirt	Two electrodes on the back and a reference electrode at the lower chest	[69]
Chest belt	Nickel and copper-coated foam	T-shirt	Lower chest	[67]
Mattress and pillow	Nickel and copper-coated conductive foam	Pajama	Back	[71]
Neonatal mattress	Conductive textile	Polyurethane	Back	[72]

**Table 2 sensors-21-04174-t002:** Contact-based textile electrodes for ECG sensor.

Integrated in	Materials of Electrodes	Sheet Resistance (Ω/sq)	Location of Electrodes	Lead Configuration	Electrode Size (cm^2^)	References
Vest with Velcro Strap	Nickel/copper-coated foam	0.07	Chest	Lead II	8	[102]
Elastic band	Silver-plated knitted fabric	<100	Lower chest	Lead I	8, 4.5, 2.25	[83]
Chest belt	Ag/Ti-coated yarn embroidered into the fabric	NK	Chest	Lead I	14	[103]
Plastic clamps	PEDOT:PSS screen printed cotton, cotton/lycra	22.7–117.3 ^a^	Forearms and ground on the ankle	Lead I	10	[104]
Elastic band	Silver-plated nylon	<1	Wrist and biceps	NK	NK	[29,105]
Elastic band	Polypyrrole-coated cotton	325	Forearms and ground on the ankle	NK	12.25	[106]
Chest belt	Silver-plated nylon	NK	Around abdominal	Lead I	1, 2, 4, 8, 16	[107]
Adjustable chest strap	Ag/AgCl printed nonwoven fabric	NK	Chest	Lead I	7.07, 3.14, 0.79	[108]
Chest belt	Silver-plated nylon fabric	NK	Chest	Single lead	8, 4.5, 2.25	[81]
Chest belts	Silver-plated nylon thread embroidered on polyester fabric	NK	Chest	V3, V4, and V5 lead	1.21	[109]
Arm strap	Silver and CNT/PDMS-coated PET substrate	NK	Forearms and ground on leg	Lead I	8.04, 4.5, 2	[91]
T-shirt	Ag/AgCl electro-platted textile	NK	Chest and limbs	12 Lead	NK	[110]
T-shirt	Silver-platted textile	NK	Chest and limbs	12 Lead	NK	[74]
Plastic clamps	PEDOT:PSS-coated polyamide	NK	Forearms and ground on leg	Lead I	9	[92]
Bedsheet	Silver-plated nylon thread embroidered on a bedsheet	NK	Upper body	NK	7.04	[111]

NK: not known, the information is not given in the reference; PDMS—polydimethylsiloxane, ^a^ The unit in Ref. [104] is kΩ.

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
