# Peer review of "Wearable Smart Textiles for Long-Term Electrocardiography Monitoring—A Review"

_sensors, 2021, doi:10.3390/s21124174_

Round 1

Reviewer 1 Report

Figure 2, especially b) is not very informative. In b), the description of where the skin is is still missing. Where is the capacity? Between minus and plus designation? Maybe the picture can be improved a bit.  

What does polarizable mean exactly? Is it good or bad conductivity or high or low contact resistance?

Figure 4 and 6d) has a very poor quality. Please improve if possible.

Possibly a reference to embroidered electrodes, with conductive yarns can still be found. 

Author Response

Thanks for your valuable comments which will make our paper technically sounder and scientifically stronger, we have uploaded the response in word file.

Reviewer 2 Report

Dear Authors,

in your interesting manuscript, the following points should be added/changed to further improve it:

  • Introduction: The difference between passive and active smart textiles is not clear for me here. If the smart textile reacts to a stimulus, where is the difference between an electronic component rolling up the sleeves of a temperature-aware shirt and a shape-memory material doing the same? To the best of my knowledge, passive smart textiles contain only sensors, no actors, so they should not react, but only measure environmental conditions.
  • Section 2: You mention that mostly only two electrodes or max. three are used. Are there no approaches to establish a full 12-lead system on the base of textile electrodes?
  • Fig. 1b: This person looks quite stretched; please check if the images is correct.
  • line 165: Are the Ag/AgCl electrodes examples for polarizable or non-polarizable electrodes?
  • Is there a chance to insert Fig. 4d and Fig. 6d in better quality?
  • line 477: (16.3 ± 0.1) dB, else the average doesn't have a unit.
  • Table 2: The unit of the sheet resistance is Ohm, not Ohm/sq, although the latter is often found in the literature.
  • line 522: Ag/AgCl (without the additional slash)
  • Fig. 9e looks stretched.
  • Fig. 10: I assume the last bar should start at 2016, not at 2020, and the second shuold start at 2001 to avoid double-counting.
  • Conclusion: Is there any outlook from your side, regarding the directions in which more research is necessary?

Author Response

Thanks for your valuable comments which will make our paper technically sounder and scientifically stronger, we have uploaded he response in word file.

Reviewer 3 Report

The paper presents an eminently thorough and readable review of this important field. I commend the authors' attention to the full breadth of the literature. 

I have only two relatively minor comments. The first is at line 160 ("...allow the exchange of electrons or current.") While ionic conduction can happen with ionic electrodes, the structure of this phrase made me pause, wondering if I had missed something (exchange or movement of electrons is current) . Depending on the intent of the authors, the word "current" could be replaced with "ions" or something similar. 

The other comment is about the axis of figure 10. It says 2011-2015  2020-Todate. I presume you meant 2015-Todate. But perhaps an actual date (instead of Todate) would be better since this would be an archival publication.

Thank you for your diligence in putting this excellent review together!

Author Response

Thanks for your valuable comments which will make our paper technically sounder and scientifically stronger, we have uploaded he response in word file.

This manuscript is a resubmission of an earlier submission. The following is a list of the peer review reports and author responses from that submission.

Round 1

Reviewer 1 Report

The manuscript presents a review about smart garments for ECG monitoring. The presentation is quite clear and easy to follow, even though some parts need better proofreading.

The topic was already discussed in some reviews, and some of them were cited by the authors. However, because of the previous literature in the field, the novelty of this manuscript is limited. I think the authors, when presenting their work, should make _explict_ reference to other reviews in the field rather than citing them as other research papers, by introducing them in some points of the introduction.

Moreover, some works are cited in some places but not in other where they would be pertinent (e.g., the discussion around fig. 7 about the performance of the textile electrodes during motions is marginally described. Several papers presented interesting results and need to be cited here), or some others are used to strengthen some sentences even though there is a limited significance for that citation in that point (e.g., line 106).

From a content perspective, the parts related to heart physiology, ECG signal and medical interpretations are often wrong or misleading. It seems like the authors have very little knowledge of these aspects. in my opinion, the whole section 2 should be dramatically improved or removed. Some examples:

  • line 67: premature arterial contraction? atrial?
  • lines 95-96: "... which can be measured on the skin using ECG [29,30]. This is the electrical signal generated from the sinoatrial node which is the natural pacemaker of the heart." is misleading since the ECG measured on the skin is not the signal generated from the SA node... the SA node originates the electrical activity associated to a cardiac cycle (in healthy subjects), which is completely different.
  • lines 98-99: "The full morphological structure of the ECG signal contains three major peaks called P, QRS complex, and T" is incorrect since P and T are waves, and QRS is a complex of 3 waves but they cannot be termed as "peaks"
  • line 102 "permanent and accurate heart monitoring"? Permanent?
  • line 119: "lead created at the center of two leads perpendicular to the positive electrodeposition" an electrode position cannot be perpendicular to anything. Moreover, there is a misleading use of the term "lead", which has a clear bioelectromagnetic definition
  • line 120: "(Right leg) is used as a reference electrode called the ground electrode". Really? The right leg is rarely used as ground in the ECG measurement. It is a peculiarity of ECG having this electrode usually connected to an active drive. 
  • lines 121-122 "12-lead system where unipolar and bipolar leads are obtained from the same electrodes" is absolutely wrong!
  • line 124 "with 3 electrodes lead I, II and III can be obtained." it is misleading. It depends on the position of the electrodes and, if I, II and III can be obtained, also the augmented leads can! So, it is also wrong.
  • lines 408-409 "washing, there are still no missing R-peaks, meaning the results are acceptable for ambulatory purposes." really not... the QRS detection is only a part of the story, which is not the most important one in clinical settings.

Beyond the electrophysiology, some other contents need a more rigorous treatment. For instance:

  • line 144: the authors refer to gel-based electrodes as the best examples of non-polarizable electrodes, which is misleading. Non-polarizable electrodes require a gel to improve the performance.
  • lines 142-148 should be checked since it is confusing

line 157: "this gel needs to be reapplied regularly at least in 10 hrs intervals [44]." even though there is a reference, the concept is misleading and wrong, when applied to commercial electrodes since, otherwise, they could not be used for 24/72h monitoring scenarios!

  • lines 293-298: the size of the electrodes has been studied in several works. The findings depend on the obvious reduction of the impedance when the electrode area grows. This carries on several implications that should be discussed since, for instance, the huge sensors in Fig. 3 have little sense in terms of spatial selectivity. The authors cited (in other points) several other works discussing the problem of electrode size, which could be interestingly cited again here
  • lines 299-301 the sentence is misleading and partially wrong
  • lines 385-386 the authors should explain how the direct polymerization onto a non-conductive substrate takes place.

Overall, I think that the authors should really improve the presentation and go beyond what is already know and described in previous reviews.

The quality of some figures is poor.

There are several English language issues, some reported hereafter:

  • lines 45-46 "... a patient... their..."
  • line 85 "ventricle" -> "ventricles"
  • lines 94-95 "the atrium and ventricle" should be plural
  • line 119 "electrodeposition"
  • line 161 "interfacial contact" should be renamed 
  • line 170-171 rephrase and correct
  • line 199 "heart detection" ?
  • line 199 "Textile based" please use the "-"
  • lines 211-212 rephrase and correct
  • line 217 "their weight and size should be as small and light as possible" size is not light
  • line 289 "affect"->"affects"
  • line 396 "adopted" -> "adapted"
  • line 468-469 formatting inconsistency

Reviewer 2 Report

This manuscript proposes a review on textile-based electrodes for ECG monitoring. A general introduction about smart textiles and ECG monitoring was given. However, it is better to have an in-depth presentation about the textile-based electrodes for EGC monitoring.

Wearable Smart Textile for long term electrocardiography monitoring have been developed for a long time ago. The development of the electrocardiography monitoring is not limited to the academic aspect, it is emerging in the market nearly for a half century especially for the near two decades with diversity of the product line. The writer started to investigate from the ECG signal, electrodes history, material application in the electrodes to the design application, however without any inspired novelty, argument and conclusion. The writer does not clearly state out the key problem in the area of attachment, deformation and noise when the textile electrode applied in the electrocardiogram and other electrode-related applications.  The writer is also lack of review the investigation and solution from the current and previous researcher to point the key aspect for consideration, as a result, the article is lack of readability and appropriateness.

  1. The reader is required to elaborate and analysis the research direction of a product development in the market from the previous half century as well as the problem which is people currently concerned for example, the design of the electrode on the chest, how to be positioned, how to prevent movement, and how to increase the impact in a part of the human body, how to improve the impact of skin moisture and hair, the variation of data collection which is the major concerned of the people.

  1. Before writing a thesis, the writer is required to take up some exploratory experiment to recognize the root of the problem followed by the construction of the argument. The review will become inspired; however, the rough statistics cannot create any additional value to the readers.

  1. It is suggested to elaborate more about the textile structures and constructions about the textile-based electrodes with relevant examples. it is better to address some suggestions on current problems and challenges by applying the wearable electronics for EGC monitoring. As there are a number of reviews about wearable electronics and related devices, could you please provide more advanced technologies for intensifying the significant of this review?

  1. In Table 1, ‘trousers’ instead of ‘trouser’. Please check the typo / spelling mistakes.

In summary, it is recommended reject this paper.

Round 2

Reviewer 1 Report

I think the authors improved the manuscript according to the provided suggestions even though some more changes could have been beneficial for the final result. 

I'm attaching hereafter my minor comments on the manuscript, which are strongly suggested before publication.

line 18: The adjective "comprehensive" appears redundant. The Review is well done but not comprehensive of all the works.

line 22: remove "discussed in-depth" since the discussion is rather shallow compared to other reviews

line 80: "paper of Pain [11] a survey" ->by Pani et al. [11], a survey"

lines 82-83: I don't understand these two lines here, since the manuscript discusses several technologies for the electrodes and only this one is anticipated here. I would remove the sentence (but not the reference, to be cited elsewhere)

lines 84-88: the authors should connect this part to lines 79-81 to explain why this review was needed: where is the improvement over previous works and the motivation for this new work, even simply as an update.

line 93: "sensible" -> important

line 113: SA node is not the origin of the electrical current but only the origin of the electrical activation of the heart. Otherwise, it would seem like a current source, which is not.

line 126: "around the chest and limbs" -> "around the chest and on the limbs"

line 133: "are derived from the same three electrodes as bipolar limp leads." -> "are derived from the same three electrodes used for the bipolar leads."

line 138: the authors missed to explain that the augmented leads can be computed based on the bipolar ones.

line 139: No. A fourth electrode is used to provide a ground reference, usually through active circuits.

figure 1 (a) is misleading without an indication of a lead in which the signal typically assumes that shape

line 160-161: No. Clinical ECG at rest is a 10-second recording. The duration indicated by the authors is not a clinical standard

figure 2 is misleading, especially figure 2(a) which is imprecise if not wrong

lines 263-268: I completely disagree. The capability of the electrode to provide good contact with the skin is not associated with the good conductivity since typically redox reactions occur at the interface with the electrode. For instance, the AgCl is not a very good conductor but an excellent material for the electrodes.

line 337: "adopted" -> "adapted"?

lines 350-352: The sentence sounds bad. I think something is missing. Do you mean "An ECG electrode developed through silver paste screen printing on cotton and polyester fabric was reported in [24], and it was reported that 15 mmHg pressure is required to avoid excessive motion artefacts. " ?

line 355: "due to increase the contact" -> "due to increase in the contact"

line 365: that is not the only effect of reduced skin contact impedance.

line 482: I think that [95] should be added to the list. Moreover, the discussion around [23] is affected by the limits of that work in the positioning of the electrodes which should be never adopted for dynamic recording. A better discussion on the topic is provided in [95]. The authors can keep the present condition only explaining the methodological limit of that work.

line 573: "rare" -> "rate"

Reviewer 2 Report

Author amended the paper, still I am afraid that the significance for research regarding wearable sensor from this review paper is not able to be found. This paper cannot present the current problem and new methodologies of wearable sensors.